# NLRP3 and Gut Microbiota Homeostasis: Progress in Research

**DOI:** 10.3390/cells11233758

**Published:** 2022-11-24

**Authors:** Hongming Pan, Yuting Jian, Feijie Wang, Shaokun Yu, Jiannan Guo, Juntao Kan, Wei Guo

**Affiliations:** 1Department of Pharmacology, School of Pharmacy, Fudan University, Shanghai 201203, China; 2Department of Gastrointestinal Surgery, Harbin Medical University Cancer Hospital, Harbin Medical University, Harbin 150081, China; 3Nutrilite Health Institute, Shanghai 201203, China

**Keywords:** NLRP3, microbiota, intestinal mucosal immunity, inflammasome

## Abstract

The inflammasome is a platform for inflammatory signaling, and the NLRP3 inflammasome recognizes stimuli in vitro and in vivo, and releases inflammatory cytokines that trigger inflammation and pyroptosis. In the gut, the NLRP3 inflammasome is a key sensor for protecting the body from damage and exogenous pathogens. It plays a fundamental role in maintaining the stability of the gut’s immune system. We focus on the role of NLRP3 as a key node in maintaining the homeostasis of gut microbiota which has not been fully highlighted in the past; gut microbiota and innate immunity, as well as the NLRP3 inflammasome, are discussed in this article.

## 1. Introduction

The study of gut microbiota focuses on clarifying the composition of normal gut microbiota and maintaining a stable, normal intestinal flora structure in the host, which is crucial in maintaining the normal intestinal function of the host and resisting pathogenic factors. At present, the main work in the research field of gut microbiota is trying to understand the types and composition ratio of human gut microbiota, and even the distribution characteristics of flora, more accurately and comprehensively in different intestinal segments, find out the differences of gut microbiota among individuals in different health states and of different ethnicities, and explore the influence of interference factors such as antibiotics and dietary habits on intestinal flora composition [1,2,3]. Based on the current understanding of gut microbiota, people rebuild the host’s disturbed gut microbiota through flora transplantation to restore the normal and stable state of the host’s gut microbiota and maintain the normal intestinal state of the host [4]. However, the way of reconstruction after disturbance is still passive and relatively lagged. We still need to further clarify the key nodes in the gut microbiota change to find a breakthrough point with application value, so that we can take greater initiative in maintaining the stability of the gut microbiota. Therefore, we focus on the role of NLRP3 as a key node in the intestinal flora change to highlight the great potential of intervention against NLRP3 in the maintenance of the gut microbiota.

## 2. Inflammasome and Innate Immunity

The concept of inflammasomes was first proposed by Jurg Tschopp in 2002, and was composed of three main components: (1) pattern-recognition receptors (PRR) or NOD-like receptors (NLR) or AIM2-like receptors (ALR); (2) apoptosis-associated speck-like proteins containing caspase recruitment domains (ASCs); and (3) caspase proteases [5]. The polyprotein complex with those three parts is appproximately 700 kDa [6]. The innate immune system protects the body from pathogenic microorganisms and harmful substances and is involved in maintaining homeostasis in the gastrointestinal tract. As an important receptor and sensor in the innate immune system, the inflammasome plays an important role in inflammatory signaling by recognizing pathogenic factors, such as bacteria, fungi, viruses, and products of cell damage, and by initiating inflammatory responses through pro-inflammatory factors [7]. The production of inflammatory factors and cell death mediated by inflammasome activation is closely related to the body’s defense against pathogenic microbial infection and the maintenance of homeostasis, making this phenomenon an integral part of innate immunity. Something like pathogen-associated molecular patterns (PAMPs) from invading pathogens, such as lipopolysaccharide (LPS), peptidoglycan, muramyl dipeptide (MDP), bacterial RNA, and danger-associated molecular patterns (DAMPs) from damaged and dying cells in the host, triggered PRR [8]. PRR binds to the adapter ASC and effector pro-caspase-1 to complete NLRP3 inflammasome assembly, then promotes pro-caspase-1 self-cleavage to switch on the production of activated caspase-1, which further triggers IL-1-family cytokines, pro-IL-1β and pro-IL-18, into biologically activated inflammatory factors, IL-1β and IL-18, through proteolysis [9]. Therefore, the inflammasome is regarded as a key factor required for the secretion of IL-1β and IL-18, which play a vital role in innate and adaptive immunity and inflammatory responses in the body [10] (Figure 1).

## 3. Gut Immunity and Gut Microbiota

The intestinal tract is not only the largest digestive organ but also an essential immune organ. Although the intestinal epithelium is directly challenged by microorganisms since it is directly exposed to stimuli from the external environment, the human intestine is at the forefront of the immune system, with over 70% of all immune cells settled here. Human intestinal mucosa covers approximately 300 m^2^ and is the main site of contact between the body and pathogens [11]. The human gut is colonized by a large number and species of gut microbiota, which form a dynamic and mutually beneficial symbiotic relationship with the human body and play a critical role in gut immunity [12]. Therefore, together with experimental epithelial cells (IEC), intraepithelial lymphocytes (IEL), lamina propria lymphocytes (LPL), and Peyer’s patches (PP), the gut microbiota account for one of the primary components of the intestinal immune system [7,8]. Many microorganisms exist in the gut microbiota, including bacteria, archaea, fungi, protozoa, and viruses [13]. Among all, intestinal bacteria, with a number of approximately 10 [14], form a complex but relatively stable huge system. Based on microbial culturomics research, these bacteria are mainly divided into Firmicutes, Bacteroidetes, Actinobacteria, Proteobacteria, and Clostridium, of which Bacteroidetes and Firmicutes together account for approximately 90% of the total gut microbiota (25% and 65% respectively) [14]. However, intestinal environments, as well as gut microbiota, are intestinal segment and individual dependent. For example, gut microbiota is influenced by diet and genetics, which is most obvious in the composition of Bacteroidetes and Proteobacteria [15].

### 3.1. Gut Microbiota Is Essential for Gut Immunity

Gut microbiota not only contributes to nutrient absorption and metabolism in the body but also competes with bacterial and fungal pathogens for nutrients and attachment sites and resists their colonization of the intestinal mucosa by producing antibacterial substances and absorbing foreign pathogenic factors [16]. Therefore, gut microbiota enables the functional gastrointestinal immune system; it is even possible to view the gut microbiome as an organ system that is important to the development of children and plays a key role in child development [17]. In the first five years of life, the intestinal flora will affect the development of the infant’s immune system and preschool reasoning ability. During this period, the healthy development of the infant’s intestinal flora will help the infant to build a perfect immune system and cognitive function [18,19]. In normal people, the gut microbiota plays an essential role in developing and functioning immune organs, such as the small intestine and intestinal lymph nodes in the ileum [17]. In recent research, the changes in the digestive tract of the body before and after gut microbiota transplantation were compared; the results confirmed that the recurrence of recurrent *Clostridioides difficile* infection (rCDI) was effectively curbed after fecal microbiota transplant (FMT) [20]. In the treatment of rCDI infection, FMT affects the life cycle of *C. difficile* by restoring the interaction between gut microbiota and metabolites, especially short-chain fatty acids (SCFA) and bile acids.

Up to now, there are sufficient clinical research data to support that FMT has become an effective treatment for *C. difficile* infection [21]. Ekekezie Chiazotam’s survey shows that the overall efficacy rate of microbiota transplantation in the treatment and prevention of *C. difficile* infections is between 80% and 90%, with no serious adverse reactions during the treatment [22]. Gut microbiota is also indispensable for developing an optimal immune system. The gut microbiota colonized in the gastrointestinal tract can help the host establish a strong intestinal mucosal protective layer to improve the resistance of the host’s digestive tract to colonization and foreign pathogen infection, so the development of intestinal microbiota has an important impact on the development of human immune system [23]. The transplantation experiment of intestinal microbiota in animal models has confirmed that intestinal microbiota can help the development of the host’s innate and adaptive immune system. It also demonstrated that the underdeveloped host’s intestinal immune system can be corrected by transplanting specific intestinal symbiotic bacteria. The rapid changes in intestinal microbiota will affect the development of innate immune cells and adaptive immune cells, thus affecting the homeostasis and immune system functions of the host’s intestine [24,25].

### 3.2. The Gut Microbiota Disorder Is Associated with the Pathogenesis of Various Diseases

A large number of studies have shown that gut microbiota disorder is closely related to the occurrence of a variety of diseases, such as obesity, diabetes, cardiovascular and cerebrovascular diseases, inflammatory bowel disease (IBD), irritable bowel syndrome (IBS), autoimmune diseases, allergies, autistic disorders, depression, and senile dementia [26,27,28]. The results of fecal community analysis showed that the bacterial species diversity of fecal communities in patients with recurrent *C. difficile* infection was significantly reduced [29]. Disruption of the gut microbiota at the colonic sites and subsequent exposure to *C. difficile* are the first steps in disease pathogenesis. Changes in gut microbiota increase the content of certain free amino acids, especially proline, an essential growth factor of *C. difficile,* closely related to the pathogenesis of diseases [30,31]. Except for providing the energy to support the growth of *C. difficile*, increased proline also affects the production of Clostridium difficile toxin A and toxin B, which are the leading causes of hospital-acquired diarrhea in adults and can even lead to death in patients with severe symptoms [30,32,33]. Changes in the composition of the gut microbiota will affect the metabolic balance in the body and, therefore, involve the pathogenesis of the cardiovascular disease [34]. Metabolomics studies and germ-free mouse experiments have demonstrated that gut microbiota plays a key role in trimethylamine N-oxide (TMAO) production. Phosphatidylcholine from eggs, milk, red meat, and other food in the daily diet is metabolized in the gut to trimethylamine by gut microbiota, and then to TMAO by liver-derived flavin-containing monooxygenase. TMAO promotes the upregulation of multiple macrophage scavenger receptors associated with atherosclerosis. Therefore, increased TMAO intake can accelerate the development of atherosclerosis, enhanced macrophage cholesterol accumulation, and foam cell formation [35]. As mentioned above, gut microbiota can not only regulate the mucosal immune function of the intestinal tract but also affect the occurrence and development of cardiovascular diseases by affecting the body’s nutrient metabolism.

## 4. The Relationship between the NLRP3 Inflammasome and Intestinal Homeostasis

### 4.1. NLRP3 Inflammasome Plays a Key Role in Intestinal Mucosal Immunity

As the natural cavity of the human body, the intestinal tract is frequently in contact with the external environment and foreign substances, thus serving as the primary way for many bacterial and viral pathogens to invade the human body. Stimulating molecules and pathogenic proteins contained in food and foreign microorganisms are the main inducers of the gut immune response; the intestinal defense system composed of intestinal microbiota, the intestinal epithelial barrier, and the local immune system can resist the stimulation of these stimulators or pathogens to the body, to maintain steady and intestinal health, despite the complex external environment [36]. As pathogen sensors, pattern-recognition receptors, such as the Toll-like receptor (TLR), NOD-like receptor (NLR), and RIG-I-like receptor (RLR), distributed in the intestinal epithelium can recognize the pathogen-associated molecular pattern (PAMP) ligand. Subsequently, downstream signaling pathways and molecular events are active. They induce the expression of anti-infective cytokines and other intestinal mucosal immune defense molecules to promote the occurrence of mucosal immune responses, which are crucial for maintaining gastrointestinal homeostasis [37]. Intestinal intraepithelial lymphocytes (IELs) and lamina propria T lymphocytes are the main immune cells of the gut, while dendritic cells are one of the professional antigen-presenting cells (APCs). As one of the most important members of the inflammasome family, the NLRP3 inflammasome is widely present in epithelial cells and immune cells. The molecular structure of NLRP3 contains a central nucleotide-binding and oligomerization (NACHT) domain which has ATPase activity, while the C-terminus of NLRP3 retains a leucine-rich repeat (LRR) domain to regulate the biological activity of NLRP3 and detects endogenous alarm proteins or microbial-derived ligands. When the NLRP3 inflammasome is activated by pathogens, such as bacterial toxins, ATP, reactive oxygen species (ROS), in the gut and danger signal molecules in the body, its downstream caspase-1 effector proteins activate inflammatory factors such as IL-1β and IL-18, thereby triggering an inflammatory response in the gut. Hereby, NLRP3 plays a crucial role in the body’s anti-infection immunity and the occurrence of inflammatory diseases such as IBD, IBS, and other immune-related intestinal diseases [38]. Accordingly, the role of NLRP3 inflammasome in intestinal mucosal immunity has received more and more attention and is expected to be a new target for drug discovery and development [39].

### 4.2. Downstream Effector IL-1

IL-1 is a multifunctional cytokine, which is activated by the initiation of the NLRP3 inflammasome. IL-1 is dominant in innate immunity, with strong pro-inflammatory activity, and is one of the most critical medium factors in inflammation and host anti-infection response. IL-1 can induce the expression of proinflammatory cytokines or chemokines such as IL-6, IL-8, and CCL2 in inflammatory sites. The changes in the expression of these inflammatory cytokines or chemokines recruit monocytes and neutrophils to migrate to infection, injury, and necrosis sites to engulf pathogens, inducing local inflammatory response, and even promoting the activation of dendritic cells and neutrophils; this series of chain reactions triggered by IL-1 will eventually help the host to resist the invasion of pathogens and maintain the homeostasis in the host’s intestine [40,41,42]. In addition, Judy Lieberman’s research shows that NLRP3 can affect lamina propria mononuclear cells to secrete more IL-1β. It can help the intestinal epithelium to produce antibacterial peptides and enhance the activation of Treg cells in the host body’s immune response. The activated Tregs can boost the controllability and stability of intestinal inflammatory activities to keep the stability of intestinal links; IL-1 also co-mediates the differentiation of CD4 + T cells into Th17 cells with other cytokines [9,43]. The role of IL-1 in intestinal inflammation has been reported in several studies; as a typical pro-inflammatory cytokine IL-1β occupied a key position in inducing the increase in the epithelial tight junction permeability in the intestine, which plays an important role in amplifying the cascade of intestinal inflammation. A defective intestinal epithelial tight junction (TJ) barrier is recognized as an important contributor to intestinal inflammatory diseases; blocking IL-1 can improve the status of patients with Clostridium- and Salmonella-induced colitis. These findings suggest that IL-1 is a key pro-inflammatory cytokine in IBD-like diseases which plays a core role in promoting inflammatory damage [44,45].

### 4.3. Downstream Effector IL-18

IL-18 not only activates monocytes but also stimulates the secretion of IFN-γ by Th1 cells, natural killer (NK) cells, and natural killer T cells; it also promotes the production of γδT cells which then secrete IL-17 when IL-18 is combined with IL-12, and this series of chain reactions activated by IL-18 helps the human body eliminate infectious microorganisms [46]. Moreover, IL-18 helps the body defend against bacterial and viral invasion and maintains intestinal microbiome homeostasis by activating intestinal epithelial cells to produce antimicrobial peptides. IL-18-activated γδT cells play an important role in intestinal mucosal immunity by maintaining intestinal epithelial homeostasis. Evidence shows that the combination of IL-18 and IL-2 activates and induces the proliferation of human CD56 + CD11c + precursor NK cells and consequently triggers strong proliferation and activation of γδT cells in the gastrointestinal inflamed mucosa of patients with ulcerative colitis [47]. By activating γδT cells, IL-18 also drives the production of IL-17. It is crucial for resistance against pathogen invasion and maintenance of the steady state of the intestinal tract by recruiting neutrophils to the site of pathogen infection or inflammation [48,49]. In general, intestinal epithelial cells without intestinal infection and inflammation under physiological conditions are the main source of IL-18; IL-18 released by intestinal epithelial cells has the function of promoting intestinal epithelial repair, renewal, and maturation. Therefore, it helps maintain the normal function of the intestinal epithelial barrier against intestinal infection and inflammation. More IL-18 is produced downstream of the activated inflammasome to exert its biological functions when gastrointestinal infection and intestinal inflammation occur [50].

### 4.4. NLRP3 Inflammasome and Gasdermin D

A recent study reported that caspase-1, activated by the NLRP3 inflammasome, can cleave and activate gasdermin D protein (GSDMD), which is a key mediator of innate immune defense. The study from Feng Shao’s group confirmed that GSDMD is a direct-acting molecule of inflammation-induced pyroptosis, leading to inflammatory form cell death or pyroptosis [51]. This new cell death mode which is known as “inflammatory programmed cell death” has confirmed that inflammasomes play a role in promoting the host to resist the invasion of microbial pathogens and producing aseptic inflammation [52]. The molecular structure of gasdermins contains a cytotoxic N-terminal domain and a C-terminal repressor domain connected by a flexible linker. After being cleaved by caspase, the N-terminal domain fragment of GSDMD is released and oligomerized to form a bioactive p30 fragment with a molecular weight of 32 kD; p30 then binds to the cell membrane to form many honeycomb-like pores with an inner diameter of 10–14 nm, which further promotes the release of mature IL-1 β and IL-18 [53,54]. IL-1 β and IL-18 then activate an inflammatory response in the host’s body. Furthermore, membrane pores formed by GsdmD p30 allow the inflow and outflow of various ions, proteins, and water, which cause cell osmotic pressure imbalance and cell swelling, eventually leading to inflammatory cell death or pyroptosis [55]. This phenomenon has been observed in a variety of physiological and pathophysiological conditions, from viral and bacterial infections to inflammation-related diseases, as shown in Figure 2.

Pyroptosis induced by the activation of immune response after the recognition of risk factors by inflammasomes destroys the cells infected by pathogens; the replication of pathogens in the cells is inhibited in this way to avoid the further spread of pathogens in the intestine [56]. At the same time, IL-1-family cytokines, IL-1β and IL-18, that communicate with other cells will also be released to further activate the inflammatory activity and immune response against related pathogens, thus stimulating the immune system to eliminate pathogenic infection factors and helping maintain the stability of the gastrointestinal tract [57]. Although pyroptosis protects the body from pathogenic infection by antagonizing infection or endogenous dangers, hyperactivation of pyroptosis can lead to fatal sepsis [58]. Loss of GSDMD impairs the body’s ability to clear the intestinal bacterial infection and enteroviral infection. However, the inflammasome has a complex mode of action in the intestine. Its detailed mechanism of action in intestinal immunity still needs more in-depth research.

## 5. Conclusions

After a long co-evolution with the host, the gut microbiota formed a complex but effective intestinal immune system, helping the host resist the invasion of pathogens. Many studies have confirmed that the inflammasome plays a key role in intestinal immunity, but the detailed mechanism of the inflammasome in the intestinal immune system still needs more in-depth research. The discovery of the gasdermin family provides new directions and clues for the study of inflammasomes in intestinal immunity. However, its application as a potential therapeutic target for practical clinical therapeutic applications still needs further exploration [59]. The research on intestinal mucosal immunity has attracted increasing attention; as an important part of the body’s immune defense line, intestinal mucosal immunity is indispensable for the maintenance of the body’s homeostasis. Therefore, studying the mechanism and pathways of the intestinal immune system is of great significance for related diseases such as inflammatory bowel disease. The existing research shows that NLRP3 plays a key role in intestinal flora homeostasis and intestinal mucosal immunity. As a signaling molecule in the intestinal inflammatory response, NLRP3 provides new ideas to understand and improve the therapeutic effect of IBD. With the deepening of research work, NLRP3 may become a therapeutic target for intervening intestinal diseases such as inflammatory bowel disease and intestinal infection, but more clinical studies are needed before practical application.

## Figures and Tables

**Figure 1 cells-11-03758-f001:**
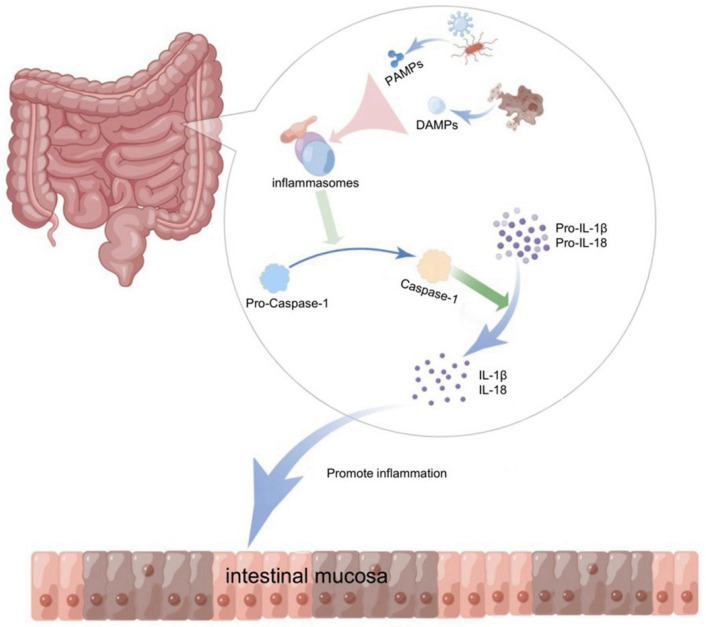
Activated inflammasomes in the gut recognize stimuli and initiate inflammatory responses. Inflammasomes in the intestine recognize the pathogen-associated molecular patterns (PAMPS) of exogenous pathogens invading the host, or the damaged and dying cells release the dangerous-associated molecular patterns (DAMPs) in the body, which will stimulate ASC to assemble with pro-caspase-1 to form inflammasomes. The assembled inflammasomes promote pro-caspase-1 to self-cleavage and produce activated caspase-1, and the activated caspase-1 will convert pro-IL-1 β and pro-IL-18 to IL-1 β and IL-18, which in turn triggers an inflammatory response in the intestine (diagram by Figdraw).

**Figure 2 cells-11-03758-f002:**
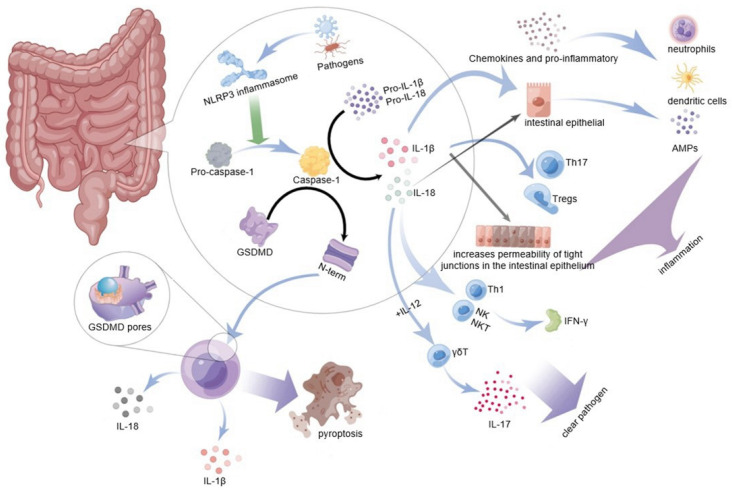
NLRP3 inflammasome in the intestine recognizes bacteria, viruses, or other pathogens and promotes the activation of pro-cassase-1 to caspase-1. On the one hand, caspase-1 will cleave GSDMD to produce an active segment N-terminal, which can bind to the cell membrane to form many 10–14 nm holes to cause pyroptosis and release IL-1 β and IL-18 to promote inflammatory reaction. On the other hand, activated caspase-1 activates pro-IL-1 β and pro-IL-18, activated IL-1 β, further inducing the expression of inflammatory cytokines and chemokines in the inflammatory part; it can promote the recruitment and activation of neutrophils and dendritic cells in the inflammatory part, and promote the differentiation of Th17 and Tregs. IL-18 can activate intestinal epithelial cells to produce AMPs, stimulate the production of Th1, NK, and NKT to secrete IFN- γ, or combine with IL-12 to induce more γδ T cells which secrete IL-17. NLRP3 inflammasome promotes the local inflammatory response of the intestinal and helps remove the pathogenic factors invading the body to maintain intestinal homeostasis through the above way (diagram by Figdraw).

## Data Availability

Not applicable.

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
