# Peer review of "NLRP3 and Gut Microbiota Homeostasis: Progress in Research"

_cells, 2022, doi:10.3390/cells11233758_

Round 1

Reviewer 1 Report

The review presented by Pan et al., aims to analyse the role of NLRP3 as a key node in maintaining the homeostasis of gut microbiota. The topic is relevant and the article is well-structure. Only minor issues need to be changed:

Introduction is tood lax and short. The current state of the research field should be reviewed carefully. You should mention the main aim of the work and highlight the main conclusions.

Review: C. difficile must be in italic along the text

Review: Funding and Institutional Review Board Statement. They are incomplete

Author Response

Reply: We thank you for this valuable comment. We write a point-by-point response to the reviewer’s comments as follows:

  1. The introduction added the current situation in the field of gut microbiota research, also mentioned the main objectives of this paper and emphasized the main conclusions of this paper. We highlight the rewrite part of our draft. Accordingly, we have added four corresponding references. We have mentioned all the Figures as you suggested. 
  2. C. difficile has been changed to C. difficile
  3. Funding and Institutional Review Board Statement has been supplemented.

Reviewer 2 Report

After a long co-evolution with the host, the gut microbiota has formed a complex  but effective intestinal immune system, helping the host resist the invasion of pathogens.

Many studies have confirmed that the inflammasome plays a key role in intestinal immunity, but the detailed mechanism of the inflammasome in the intestinal immune system still needs more in-depth research.

The discovery of the Gasdermin family provides new directions and clues for the study of inflammasomes in intestinal immunity, However, its application as a potential therapeutic target for practical clinical therapeutic applications still needs further exploration

The research on intestinal mucosal immunity has attracted increasing attention; as an important part of the body's immune defense line, intestinal mucosal immunity is indispensable for the maintenance of the body's homeostasis.

Therefore, studying the mechanism and pathways of the intestinal immune system is of great significance for related diseases such as inflammatory bowel disease. The existing research shows that NLRP3 plays a key role in intestinal flora homeostasis and intestinal mucosal immunity.

As a signaling molecule in the intestinal inflammatory response, NLRP3 provides new ideas to understand and improve the therapeutic effect of IBD. With  the deepening of research work, NLRP3 may become a therapeutic target for intervening intestinal diseases such as inflammatory bowel disease and intestinal infection, but more clinical studies are needed before practical application.  

Author Response

Reply: Thanks for the reviewer’s good suggestion. We have mentioned the draft as you suggested and by highlights the revised part. 

Answer to Q1: There are many research results proving that the inflammasome plays an important role in gastrointestinal inflammatory activities. It seems to be a key node connecting a series of inflammatory activities in the gastrointestinal tract to form a huge and complex network. Although there is no mature clinical intervention plan targeting NLRP3 at this stage, more clinical research and exploration are needed. However, it is a more proactive strategy to intervene in the early stage of gastrointestinal inflammation and other stress reactions through the regulation of NLRP3, to maintain the homeostasis of the intestinal environment in time before significant changes in the homeostasis of gut microbiota. It is worth exploring more clinical research work.

Answer to Q 2:In this paper, we present the role of inflammasomes as signal sensors and critical nodes, which connect inflammation and other stress reactions in the gastrointestinal tract to form a huge and complex system. Inflammasome deserves more exploration to clarify its value in maintaining gut microbiota homeostasis. 

Answer to Q3: At this stage, the research on gut microbiota focuses on the in-depth exploration of the composition of gut microbiota and attempts to reconstruct gut microbiota by means of flora transplantation technology. However, such a reconstruction method has hysteresis and passivity, it is difficult to prevent damage in advance. Therefore, exploring the role of the inflammasome in the key nodes of gastrointestinal inflammatory activities and the correlation with intestinal flora homeostasis will help us understand the importance of provocative bodies in intestinal flora homeostasis, However, more clinical studies are needed to verify the therapeutic measures targeting NLRP3.
